# Computer-aided X-ray screening for tuberculosis and HIV testing among adults with cough in Malawi (the PROSPECT study): A randomised trial and cost-effectiveness analysis

Peter MacPherson[1,2,3]*, Emily L. Webb[4], Wala Kamchedzera[2], Elizabeth Joekes[1], Gugu Mjoli[5], David G. Lalloo[1], Titus H. Divala[2,3,6], Augustine T. Choko[1,2], Rachael M. Burke[2,3], Hendramoorthy Maheswaran[7], Madhukar Pai[8], S. Bertel Squire[1], Marriott Nliwasa[2,6], Elizabeth L. Corbett[2,3]

1 Department of Clinical Sciences, Liverpool School of Tropical Medicine, Liverpool, United Kingdom, 2 Malawi-Liverpool-Wellcome Trust Clinical Research Programme, Blantyre, Malawi, 3 Clinical Research Department, London School of Hygiene and Tropical Medicine, London, United Kingdom, 4 MRC Tropical Epidemiology Group, London School of Hygiene and Tropical Medicine, London, United Kingdom, 5 Department of Radiology, Chris Hani Baragwanath Hospital, Soweto, South Africa, 6 Helse Nord TB Initiative, College of Medicine, University of Malawi, Blantyre, Malawi, 7 Department of Public Health and Policy, University of Liverpool, Liverpool, United Kingdom, 8 McGill International TB Centre, McGill University, Montreal, Canada

* peter.macpherson@lstmed.ac.uk

## Abstract

### Background

Suboptimal tuberculosis (TB) diagnostics and HIV contribute to the high global burden of TB. We investigated costs and yield from systematic HIV-TB screening, including computer-aided digital chest X-ray (DCXR-CAD).

### Methods and findings

In this open, three-arm randomised trial, adults (≥18 years) with cough attending acute primary services in Malawi were randomised (1:1:1) to standard of care (SOC); oral HIV testing (HIV screening) and linkage to care; or HIV testing and linkage to care plus DCXR-CAD with sputum Xpert for high CAD4TBv5 scores (HIV-TB screening). Participants and study staff were not blinded to intervention allocation, but investigator blinding was maintained until final analysis. The primary outcome was time to TB treatment. Secondary outcomes included proportion with same-day TB treatment; prevalence of undiagnosed/untreated bacteriologically confirmed TB on day 56; and undiagnosed/untreated HIV. Analysis was done on an intention-to-treat basis. Cost-effectiveness analysis used a health-provider perspective. Between 15 November 2018 and 27 November 2019, 8,236 were screened for eligibility, with 473, 492, and 497 randomly allocated to SOC, HIV, and HIV-TB screening arms; 53 (11%), 52 (9%), and 47 (9%) were lost to follow-up, respectively. At 56 days, TB

**Data Availability Statement:** All data are available from doi:10.5061/dryad.ffbg79ctb.

**Funding:** This study was funded by Wellcome (grant: 206575/Z/17/Z to PM). The funders had no role in study design, data collection and analysis, decision to publish, or preparation of the manuscript.

**Competing interests:** I have read the journal's policy and the authors of this manuscript have the following competing interests: MP is a member of the Editorial Board of PLOS Medicine.

**Abbreviations:** AMD, adjusted mean difference; ART, antiretroviral therapy; DCXR-CAD, computer-aided digital chest X-ray; GDP, gross domestic product; ICER, incremental cost-effectiveness ratio; LF-LAM, lateral flow urine lipoarabinomannan assay; QALY, quality-adjusted life year; RR, risk ratio; SOC, standard of care; TB, tuberculosis.

treatment had been started in 5 (1.1%) SOC, 8 (1.6%) HIV screening, and 15 (3.0%) HIV-TB screening participants. Median (IQR) time to TB treatment was 11 (6.5 to 38), 6 (1 to 22), and 1 (0 to 3) days (hazard ratio for HIV-TB versus SOC: 2.86, 1.04 to 7.87), with same-day treatment of 0/5 (0%) SOC, 1/8 (12.5%) HIV, and 6/15 (40.0%) HIV-TB screening arm TB patients ($p = 0.03$). At day 56, 2 SOC (0.5%), 4 HIV (1.0%), and 2 HIV-TB (0.5%) participants had undiagnosed microbiologically confirmed TB. HIV screening reduced the proportion with undiagnosed or untreated HIV from 10 (2.7%) in the SOC arm to 2 (0.5%) in the HIV screening arm (risk ratio [RR]: 0.18, 0.04 to 0.83), and 1 (0.2%) in the HIV-TB screening arm (RR: 0.09, 0.01 to 0.71). Incremental costs were US$3.58 and US$19.92 per participant screened for HIV and HIV-TB; the probability of cost-effectiveness at a US$1,200/quality-adjusted life year (QALY) threshold was 83.9% and 0%. Main limitations were the lower than anticipated prevalence of TB and short participant follow-up period; cost and quality of life benefits of this screening approach may accrue over a longer time horizon.

## Conclusions

DCXR-CAD with universal HIV screening significantly increased the timeliness and completeness of HIV and TB diagnosis. If implemented at scale, this has potential to rapidly and efficiently improve TB and HIV diagnosis and treatment.

## Trial registration

clinicaltrials.gov NCT03519425.

---

## Author summary

### Why was this study done?

- Tuberculosis (TB), one of the leading infectious killers worldwide, remains challenging to diagnose in low-resource settings, and patients frequently face multiple health centre visits at high cost before TB is diagnosed and treatment started. HIV is a major risk factor for TB.

- Robust digital X-ray equipment can now be deployed at a primary care level in sub-Saharan Africa, and automated computer software packages that can interpret chest X-rays providing a probabilistic score for pulmonary TB have accuracy similar to, or greater than, expert human readers.

- We therefore set out to investigate whether offering adults with cough attending primary care in Blantyre, Malawi universal HIV testing and linkage to antiretroviral therapy (ART)—either alone or combined with computer-aided digital chest X-ray (DCXR-CAD) and subsequent sputum Xpert confirmation—could improve the timeliness and completeness of HIV and TB diagnosis and treatment compared to current standard approaches (health worker–directed TB and HIV screening).

## What did the researchers do and find?

- A total of 1,462 adults attending a primary clinic in Blantyre, Malawi with cough were randomly allocated to receive either standard of care (SOC) health worker–directed HIV-TB screening; oral HIV testing and linkage to treatment (HIV screening); or oral HIV testing and linkage to treatment with additional digital chest X-ray screening for TB interpreted by computer-aided diagnosis software (CAD4TBv5), with sputum Xpert testing for participants with a CAD4TBv5 score above 45 (HIV/TB screening). Participants were followed for 56 days to investigate initiation of TB treatment, missed TB and HIV diagnosis, and cost-effectiveness.

- Median time to TB treatment initiation was shorter (1 day) in the HIV-TB screening arm compared to the SOC arm (11 days) and HIV screening arm (6 days). HIV screening reduced undiagnosed/untreated HIV from 10 (2.7%) in the SOC arm to 2 (0.5%) in the HIV screening arm and 1 (0.2%) in the HIV-TB screening arm.

- Over the trial follow-up period (56 days), oral HIV testing and linkage to care were likely to be cost-effective, but digital chest X-ray with computer-aided interpretation was not.

## What do these findings mean?

- Digital chest X-ray screening with computer-aided interpretation for TB with universal HIV screening increased the timeliness and completeness of HIV and TB diagnosis.

- If implemented at scale, these interventions have the potential to rapidly and efficiently improve TB and HIV diagnosis and treatment.

## Introduction

Despite being a leading infectious cause of adult mortality worldwide [1], tuberculosis (TB) remains challenging to diagnose, especially in low-resource settings [2]. In sub-Saharan Africa, where prevalence of active pulmonary TB can exceed 1% in some high HIV prevalence urban settlements [3], adults with TB symptoms frequently make multiple visits to primary health-care services before TB testing is initiated [4]. Late and missed TB diagnosis is common, with severe consequences including hospitalisation, death, ongoing transmission, and catastrophic household expenditure associated with care-seeking and illness [4,5].

Guidelines recommend that adults attending health facilities be routinely screened for TB and investigated appropriately if symptomatic [6]. However, only a small percentage of clinic attenders successfully complete the TB screening cascade due to limited health worker numbers, high numbers of patient with symptoms of TB overwhelming testing capacity, and low availability and high cost of diagnostics [7].

Currently available TB diagnostics are not well suited to primary healthcare, and, consequently, patients are most likely to be lost during the diagnostic phase. Sputum smear microscopy has low sensitivity and is resource intensive [8]; Xpert—although more sensitive, especially for HIV–positive people [9,10]—is costly, and throughput is constrained by unit

capacity [11]; lateral flow urine lipoarabinomannan assay (LF-LAM) is not currently recommended for HIV–negative people or HIV–positive outpatients without advanced immunosuppression [12]; and sputum mycobacterial culture remains costly, slow, and inaccessible to most primary clinics. Offering newer TB diagnostics such as Xpert to all adult primary clinic attenders with symptoms of TB (which can approach 60% [7]) could rapidly overwhelm clinic testing capacity and health systems' budgets.

Rapid advances in digital X-ray technologies linked to computer-aided chest X-ray interpretation (DCXR-CAD) [13,14] mean that a "triage testing" approach to TB screening in primary care could remove barriers to high coverage of same-day, same-clinic TB diagnosis and treatment [15]. Digital chest X-ray provides a high-sensitivity and high-throughput initial screen for individuals with symptoms of TB [16], while computer-aided interpretation—software algorithms that provide an immediate probabilistic score for TB—remove the need for trained interpreters [17,18]. As specificity of chest X-ray for pulmonary TB is low, a positive X-ray triage screen should be confirmed with a high-specificity diagnostic test such as Xpert. Triage testing with DCXR-CAD could then substantially reduce the number of expensive Xpert tests that would have otherwise been required [19]. WHO has recently recommended that DCXR-CAD can be used for TB screening [18], but the impact on patient outcomes is unknown.

In this three-arm pragmatic randomised trial conducted in primary care in urban Blantyre, Malawi, we investigated whether a universal HIV testing and linkage to antiretroviral therapy (ART) intervention—either alone or combined with DCXR-CAD and subsequent sputum Xpert confirmation—could improve the timeliness and completeness of HIV and TB diagnosis and treatment. We additionally evaluated cost-effectiveness of implementation.

## Methods

### Study design

We conducted a three-arm, open, pragmatic randomised trial among adults recruited from an urban primary healthcare clinic in Blantyre, Malawi [20] (S1 Text). Bangwe Health Centre is located within a densely populated neighbourhood in the east of the city. Adult HIV prevalence in Blantyre is estimated to be 18% [21], and Malawi urban TB prevalence was estimated at 988 per 100,000 in the most recent national TB prevalence survey [22]. Comprehensive HIV care (including ART) and TB treatment are available at a primary care level through the national HIV and TB treatment programmes in Malawi.

All Blantyre-resident adults (≥18 years old) attending Bangwe Health Centre acute department who reported cough of any duration were eligible to participate and were screened for eligibility by study research assistants (6 in total, not medically trained, but with experience in community and clinic-based research) at the clinic registration desk. We excluded individuals who had taken TB treatment in the preceding 6 months, who were currently taking TB preventive therapy, or who planned to move out of Blantyre. All participants gave written (or, if illiterate, witnessed thumbprint) informed consent to participate. Ethical approval was granted by the College of Medicine, University of Malawi Research Ethics Committee, and the Liverpool School of Tropical Medicine.

### Randomisation and masking

Participants were individually randomised without restriction to 1 of 3 arms—standard of care (SOC), HIV screening, or HIV-TB screening—in a 1:1:1 ratio using a computer-generated random number sequence running on study data collection electronic devices. Because of the nature of interventions, it was not possible to blind participants or study teams to allocation.

However, investigator masking was maintained until after final statistical analysis. To minimise the risk of contamination between arms, a digital thumbprint (Simprints, Cambridge, United Kingdom) was recorded at recruitment and used to validate identity.

## Procedures

All participants underwent a baseline questionnaire by study research assistants, conducted in a dedicated study research building within the grounds of the clinic.

Participants allocated to the SOC arm were directed to the clinic acute waiting area, where they were managed by clinic health workers without any further input from study staff. As this was a pragmatic trial, we intended that the SOC arm procedures would reflect the routine TB and HIV screening care delivered by a typical Ministry of Health primary care clinic in a low-resource setting. Available to participants in the SOC arm (if requested by a clinic health worker) were the following: sputum smear microscopy; sputum Xpert, TB treatment; HIV testing using rapid fingerprick diagnosis kits; and ART, as well as other routine primary health services. Malawi National Guidelines recommend HIV testing for all adults attending health facilities, although this is often not completed [23]. Clinic health workers included nurses, clinical officers, and HIV counsellors employed by the Ministry of Health of Malawi, but no physicians.

Participants allocated to the HIV screening arm who self-reported being HIV negative, or HIV positive but not taking ART, were offered HIV testing using oral kits (OraQuick HIV 1/2 Rapid Antibody Test, OraSure Technologies, Manufactured in Thailand), with confirmatory testing using serial testing with rapid kits (Determine HIV-1/2, Alere, Japan; Uni-gold HIV, Trinity Biotech, Ireland). HIV–positive participants were supported to register for ART at the onsite clinic by walking with them to the ART clinic registration area and making an appointment for treatment initiation assessment by the ART clinic staff (in practice usually done on the same day, in accordance with Malawi guidelines). Further investigations and management, including TB investigations, were provided by clinic health workers in accordance with national guidelines without any further input from study staff.

In addition to the HIV testing intervention, participants in the HIV-TB screening arm were offered a single posterior–anterior digital chest X-ray (MinXray Commander CMDR. T.120.60. S, United States of America) done by study radiographers with interpretation using CAD4TB v5 (Delft Imaging, the Netherlands). CAD4TB interpretation was done locally on a dedicated computer, with results automatically transferred to study data collection devices. Where the CAD4TB score was $\geq$45 (threshold selected based on pilot studies and in discussion with the software developer), participants' sputum was tested onsite by Xpert; if positive, they were initiated on TB treatment by the clinic TB officer. Where the Xpert was negative or CAD4TB score was <45, they were directed to the clinic waiting area for further management by clinic health workers. Digital chest X-rays were additionally read remotely by a consultant radiologist, and participants with abnormal findings identified were recalled by telephone and referred to the clinic or city central hospital with results.

Participants were seen at the study clinic at 56 days, when they underwent questionnaire and inspection of health records to determine TB treatment status, HIV testing (if not known to be HIV positive and taking ART) and sputum collection for Xpert, MGIT culture and smear, with samples analysed at the TB Research Laboratory at the College of Medicine, University of Malawi. Participants who did not attend the clinic for this outcome assessment were

traced to home, where possible. Loss to follow-up was defined as participants who did not attend their clinic day 56 outcome assessment appointment and could not be traced to home.

## Outcomes

The primary outcome was time in days—up to, but not including, day 56—to TB treatment initiation. Secondary outcomes were the proportion of participants with same-day TB treatment initiation; proportion of participants with undiagnosed/untreated microbiologically confirmed pulmonary TB on day 56 (either sputum culture, or sputum Xpert, or sputum smear microscopy positive on day 56 sample); proportion with undiagnosed/untreated HIV; time in days—up to, but not including, day 56—to ART initiation among participants with positive confirmatory HIV test results at day 56 and who were not taking ART at day 0; mortality; and quality of life (assessed by difference in EuroQoL EQ5D utility score, a continuous variable that can take values between less than 0 and 1).

## Statistical analysis

The statistical power was estimated for the primary outcome under the assumption that 17% of adults with TB symptoms in primary care in Blantyre would initiate TB treatment by 56 days [24] (S2 Text). A sample size of 1,455 participants (485 per arm) gave at least 80% power to detect a hazard ratio (HR) for TB treatment initiation of 1.5 comparing the HIV screening arm to SOC, a HR of 1.5 comparing HIV-TB screening arm to SOC, and a HR of 1.41 comparing the HIV-TB screening arm to HIV screening arm, at 5% significance level, allowing for 5% loss to follow-up and with no adjustment for multiplicity of testing.

This study is reported as per the Consolidated Standards of Reporting Trials (CONSORT) guideline (S3 Text). We performed analysis according to the intention to treat principle. To maintain investigator blinding, no inspection of data by trial arm was done until data cleaning was complete and the database locked; the trial statistician then undertook unblinded analysis of the locked trial database. Data are deposited in the Dryad repository doi: 10.5061/dryad. ffbg79ctb [25]. Baseline characteristics were summarised for the 3 trial arms using proportions, means (with standard deviations), and medians (with interquartile ranges (IQRs)). For analysis of the primary outcome, we compared survival times using log-rank tests and constructed Cox proportional hazard regression models to estimate HRs and 95% confidence intervals (CIs) for each pairwise comparison between arms. Participants lost to follow-up were censored at the last point of contact. Log–log plots were examined and tests of Schoenfeld residuals conducted to check the proportional hazards assumption. To analyse binary secondary outcomes (proportion with same-day TB treatment initiation, proportion with undiagnosed/untreated pulmonary TB, proportion with undiagnosed/untreated HIV, and proportion reported to have died), we constructed binomial regression models with a log link function to estimate relative risk ratios (RRs) and 95% CIs, comparing between pairs of arms. To evaluate the effect of interventions on health-related quality of life, we used linear regression to compare the mean EQ5D utility scores measured at day 56 between pairs of arms, adjusting for participants' corresponding values measured at baseline. Residual plots were examined to check linear regression assumptions. For secondary outcomes with missing data, sensitivity analysis was conducted using multiple imputation with chained equations and 50 imputations.

## Cost-effectiveness analysis

Cost–utility analysis was undertaken to estimate the incremental cost per quality-adjusted life year (QALY) gained from the Malawian Ministry of Health perspective (S4 Text). Questionnaires captured the health resources utilised by participants over the trial time horizon. This

included any additional care received in hospitals and primary health clinics. Participant responses to the Chichewa version of the EQ-5D-3L were used to generate health utility scores and QALY profiles [26–28]. Incremental cost-effectiveness ratios (ICERs) were estimated to compare the 2 intervention arms to the SOC arm. Malawi does not have formal cost-effectiveness thresholds. We therefore compared the estimated ICERs to WHO-recommended thresholds using the gross domestic product (GDP) per capita for the country. Interventions that have an incremental cost per gain in QALY less than the national GDP per capita were defined as "very cost-effective," and those less than 3 times GDP per capita as "cost effective" [29]. The GDP per capital of Malawi is approximately US$400 per capital. We therefore used 1,000 bootstrapped replications to present the probability of the 2 interventions (HIV screening and HIV-TB screening) being cost-effective at increasing cost-effectiveness thresholds: US$400/QALY, US$800/QALY, and US$1,200/QALY. The probability represents the proportion of the 1,000 bootstrapped replications where the estimated ICER was below these cost-effectiveness thresholds. This analysis is reported as per the Consolidated Health Economic Evaluation Reporting Standards (CHEERS) guideline [30] (S5 Text).

## Results

Between 15 November 2018 and 22 November 2019, 8,236 patients were screened, of whom 1,462 were randomly allocated to the SOC (473), HIV screening (492), or HIV-TB screening (497) arms (Fig 1). One participant withdrew consent after baseline interview, but before randomisation.

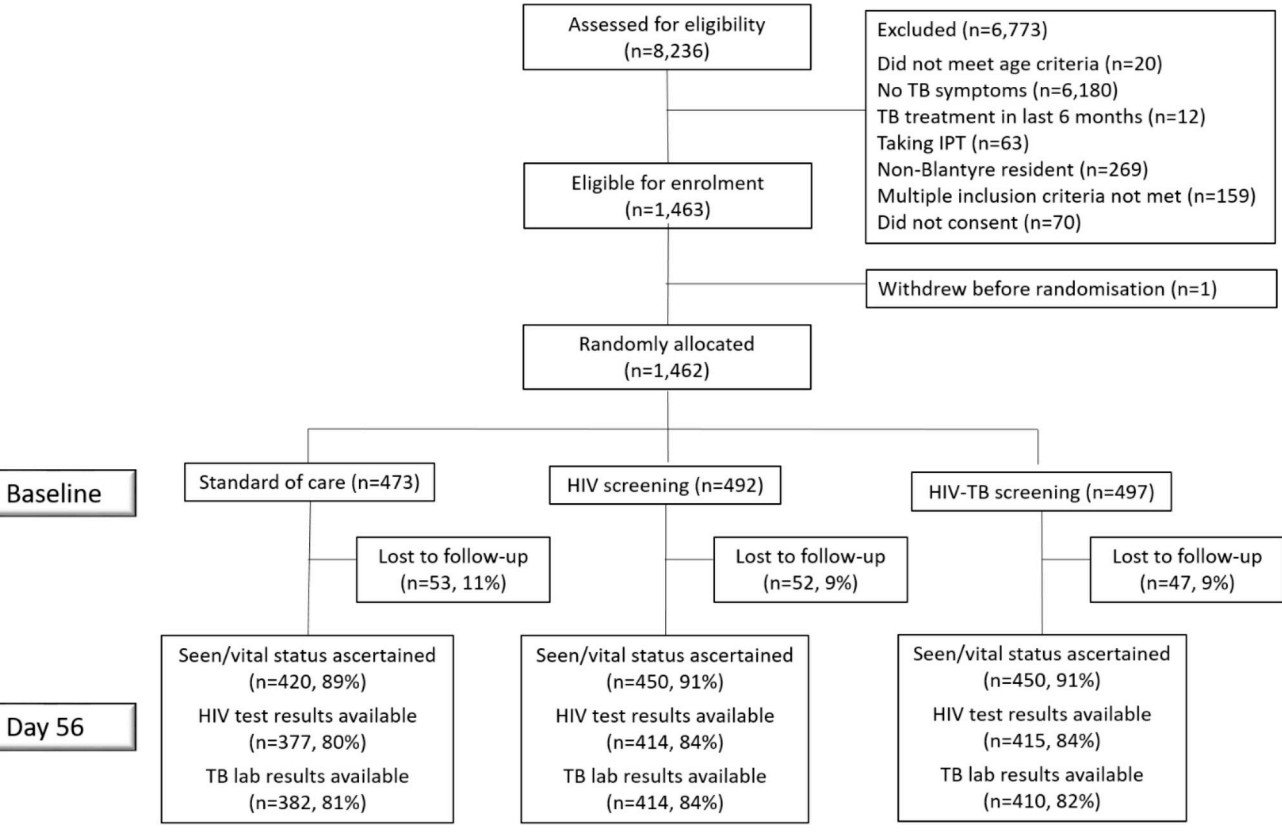

**Fig 1. Trial profile.** IPT, isoniazid preventive therapy; TB, tuberculosis.

**Table 1. Participant characteristics.**

| Characteristic | SOC arm (*n* = 473) | HIV screening arm (*n* = 492) | HIV-TB screening arm (*n* = 497) |
|---|---|---|---|
| Age in years (mean, SD) | 33.5 (13.5) | 32.8 (13.4) | 34.3 (13.4) |
| Sex | | | |
| Male (n, %) | 208 (44%) | 196 (40%) | 228 (46%) |
| Female (n, %) | 265 (56%) | 296 (60%) | 269 (54%) |
| Body mass index (mean kg/m², SD) | 22.8 (4.1) | 22.7 (4.3) | 22.9 (4.5) |
| Marital status | | | |
| Married/cohabiting (n, %) | 322 (68%) | 347 (71%) | 330 (66%) |
| Never married (n, %) | 76 (16%) | 74 (15%) | 82 (16%) |
| Widowed/separated/divorced (n, %) | 74 (16%) | 71 (14%) | 85 (17%) |
| Highest level of education | | | |
| No schooling (n, %) | 59 (12%) | 59 (12%) | 61 (12%) |
| Primary (n, %) | 217 (46%) | 234 (48%) | 234 (47%) |
| Secondary no MSCE[†] (n, %) | 130 (27%) | 131 (27%) | 125 (25%) |
| Secondary with MSCE[†] (n, %) | 60 (13%) | 56 (11%) | 70 (14%) |
| Higher (n, %) | 7 (1%) | 12 (2%) | 7 (1%) |
| Ever lost a spouse to death | 44 (9%) | 40 (8%) | 46 (9%) |
| Literate | 405 (86%) | 426 (87%) | 425 (86%) |
| Poverty quintile[ᵉ] | | | |
| uintile 1 (least poor) | 94 (20%) | 99 (20%) | 99 (20%) |
| Quintile 2 | 99 (21%) | 106 (22%) | 87 (18%) |
| Quintile 3 | 84 (18%) | 102 (21%) | 107 (22%) |
| Quintile 4 | 106 (22%) | 90 (18%) | 96 (19%) |
| Quintile 5 (poorest) | 90 (19%) | 95 (19%) | 108 (22%) |
| TB symptoms | | | |
| Cough (n %) | 473 (100%) | 492 (100%) | 497 (100%) |
| Cough duration (median weeks, IQR) | 1 (0.6, 2) | 1 (0.4, 3) | 1 (0.4, 3) |
| Night sweats (n, %) | 203 (43%) | 197 (40%) | 200 (40%) |
| Weight loss (n, %) | 194 (41%) | 186 (38%) | 195 (39%) |
| Fever (n, %) | 242 (51%) | 253 (51%) | 250 (50%) |
| Previously treated for TB (n %) | 17 (4%) | 21 (4%) | 26 (5%) |
| HIV status | | | |
| HIV–positive (n, %) | 86 (18%) | 93 (19%) | 101 (20%) |
| Taking ART (n, %) | 82 (95%) | 90 (97%) | 99 (98%) |
| HIV–negative (n, %) | 354 (75%) | 365 (74%) | 355 (71%) |
| Unknown (n, %) | 33 (7%) | 34 (7%) | 41 (8%) |
| EQ5D[§] utility score (mean, SD) | 0.79 (0.14) | 0.77 (0.15) | 0.77 (0.14) |
| Self-rated health | | | |
| Fair/good/very good | 412 (87%) | 440 (89%) | 444 (89%) |
| Poor/very poor | 61 (13%) | 52 (11%) | 53 (11%) |

[†]Malawi Secondary Certificate of Education.

[ᵉ]Based on urban proxy means test using assets derived from 2014–2015 Malawi Integrated Household Survey.

[§]EuroQOL EQ5D utility score (Zimbabwe tariff).

ART, antiretroviral therapy; IQR, interquartile range; SD, standard deviation; SOC, standard of care; TB, tuberculosis.

Characteristics of participants were balanced between arms (Table 1). Over half of participants were women in each arm, and mean ages ranged from 33 to 34 years old. Symptoms indicative of TB were common, with 40% to 43% reporting night sweats, 38% to 41% reporting

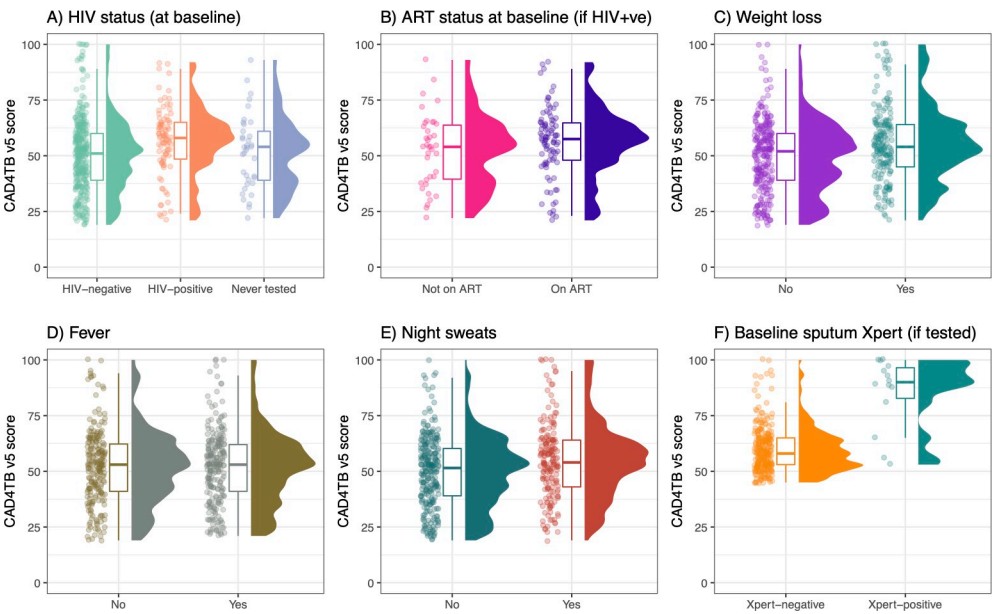

**Fig 2. CAD4TBv5 scores by participant characteristics (HIV-TB screening arm only).** ART, antiretroviral therapy; TB, tuberculosis.

weight loss, and 50% to 51% reporting fever. Between 4% and 5% of participants reported having been previously treated for TB. Knowledge of HIV status was very high (over 90%), with 18%, 19%, and 20% of participants in the SOC, HIV, and HIV-TB screening arms, respectively, self-reporting HIV–positive status. ART coverage was also very high at 95%, 97%, and 98%.

In the HIV-TB screening arm, 448 (90%) participants completed DCXR-CAD, of whom 305 (68%) had a CAD4TB score above the threshold of ≥45. CAD4TBv5 scores were similar when compared by participant characteristics (Fig 2A–2E). Of these, 279 (91%) successfully completed sputum Xpert examination. Approximately 4% (12/279) had TB confirmed on this initial sample, all of whom subsequently initiated TB treatment. CAD4TBv5 scores were higher among participants with Xpert-confirmed TB compared to participants with sputum-negative Xpert tests (Fig 2F). One further participant in the HIV-TB screening arm initiated TB treatment on day 48 with a positive Xpert result from the routine health system.

Of the 399 participants in the HIV screening arm and 396 participants in the HIV-TB screening arm who reported never having previously tested for HIV or being HIV negative, 336 (84%) and 362 (91%), respectively, received study HIV testing on the same day. A total of 1,320 (90%) participants had day 56 vital status ascertained, with similar proportions in all 3 arms (SOC: 420/473 [89%], HIV screening: 450/492 [91%], and HIV-TB screening: 450/497 [91%]), and 1,206 (82%) had day 56 TB/HIV outcomes ascertained (Fig 1). Participants lost to follow-up were slightly younger on average than those seen at day 56 and more likely to report unknown HIV status; otherwise characteristics were comparable (S1 Table).

A greater percentage of participants in the HIV-TB screening arm (15, 3.0%, 95% CI: 1.7% to 4.9%) initiated TB treatment by day 56 compared to participants in the HIV screening arm (8, 1.6%, 95% CI: 0.7% to 3.2%) or SOC (5, 1.1%, 95% CI: 0.3% to 2.4%) (Table 2), although this trend was not statistically significant.

There was a statistically significant increased rate of TB treatment initiation among participants in the HIV-TB arm compared to SOC (HR: 2.86, 95% CI: 1.04 to 7.87, $p = 0.04$), but not for comparisons between other pairs of arms (Fig 3). Overall, 60%, 100%, and 100% of

**Table 2. Effect of interventions on trial outcomes.**

| | SOC arm | HIV screening arm | HIV-TB screening arm | HIV screening vs. SOC arm | HIV-TB screening vs. HIV screening arm | HIV-TB screening vs. SOC arm |
|---|---|---|---|---|---|---|
| **Primary outcomes** | | | | **HR, 95% CI** | **HR, 95% CI** | **HR, 95% CI** |
| Number initiating TB treatment (n, %) | 5 (1.1%) | 8 (1.6%) | 15 (3.0%) | | | |
| Time (days) to TB treatment initiation (median, IQR) | 11 (6.5–38) | 6 (1–22) | 1 (0–3) | 1.51 (0.49–4.62) | 1.89 (0.80–4.46) | 2.86 (1.04–7.87) |
| **Secondary outcomes** | | | | **RR, 95% CI** | **RR, 95% CI** | **RR, 95% CI** |
| Undiagnosed/untreated microbiologically confirmed pulmonary TB (n, %) | 2/382 (0.5%) | 4/414 (1.0%) | 2/410 (0.5%) | 1.85 (0.34–10.02) | 0.50 (0.09–2.74) | 0.93 (0.13–6.58) |
| Same-day TB treatment initiation (n/N, %) | 0 (0%) | 1 (0.2%) | 6 (1.2%) | – | – | – |
| Undiagnosed/untreated HIV (n, %) | 10/377 (2.7%) | 2/414 (0.5%) | 1/415 (0.2%) | 0.18 (0.04–0.83) | 0.50 (0.05–5.48) | 0.09 (0.01–0.71) |
| Mortality (n/N, %, OR, 95% CI) | 3/420 (0.7%) | 3/450 (0.7%) | 4/450 (0.9%) | 0.93 (0.19–4.60) | 1.33 (0.30–5.92) | 1.24 (0.28–5.53) |
| | | | | **AMD (95% CI)[†]** | **AMD (95% CI)[†]** | **AMD (95% CI)[†]** |
| EQ5D[§] utility score (mean, SD) | 0.79 (0.18) | 0.82 (0.19) | 0.81 (0.18) | 0.03 (0.004–0.05) | −0.003 (−0.03–0.02) | 0.02 (0.002–0.05) |
| | | | | 0.03 (0.01–0.05) | −0.002 (−0.03–0.02) | 0.03 (0.004–0.06) |

[†]First row shows unadjusted results; second row shows results adjusted for baseline EuroQoL EQ5D utility score.

[§]EuroQoL EQ5D utility score (Zimbabwe tariff).

AMD, average mean difference; CI, confidence interval; HR, hazard ratio; IQR, interquartile range; OR, odds ratio; RR, risk ratio; SD, standard deviation; SOC, standard of care; TB, tuberculosis.

participants who initiated TB treatment in the SOC, HIV, and HIV-TB screening arms, respectively, had microbiologically confirmed disease at TB treatment initiation. In the HIV-TB arm, 6/15 (40%) of participants treated for TB achieved same-day treatment initiation, compared to 0/5 (0%) in SOC and 1/8 (12.5%) in the HIV screening arm (Fisher's exact *p* = 0.03). At day 56 assessment, 2/382 (0.5%), 4/414 (1.0%), and 2/410 (0.5%) in the SOC, HIV, and HIV-TB screening arms, respectively, had previously undiagnosed or untreated microbiologically confirmed TB (RR for HIV-TB arm versus SOC: 0.93, 95% CI: 0.13 to 6.58, *p* = 0.94).

Only 1/415 (0.2%) participants in the HIV-TB arm who underwent HIV testing at day 56 assessment had previously undiagnosed/untreated HIV infection, compared to 10/377 (2.7%) in the SOC arm and 2/414 (0.5%) in the HIV screening arm (RR for HIV-TB arm versus SOC: 0.09, 95% CI: 0.01 to 0.71, *p* = 0.02; RR for HIV arm versus SOC: 0.18, 95% CI: 0.04 to 0.83, *p* = 0.03). Of the 20, 25, and 15 participants newly diagnosed with HIV in the SOC, HIV, and HIV-TB screening arms over the course of the study (including at the day 56 assessment), 6 (30%), 15 (60%), and 9 (60%) initiated ART before day 56, with a median (IQR) number of days to ART initiation of 5 days (1 to 36), 0 days (0 to 0), and 0 days (0 to 17). There were no significant differences between pairs of arms in all-cause mortality by day 56. A sensitivity analysis using multiple imputation to impute missing data for secondary outcomes found similar results to the primary analysis (S2 Table).

Mean EuroQoL EQ5D utility scores at day 56 were significantly higher in participants in the HIV-TB arm compared to SOC (adjusted mean difference [AMD]: 0.03, 95% CI: 0.004 to 0.06, *p* = 0.02), and in participants in the HIV screening arm compared to SOC (AMD: 0.03, 95% CI: 0.01 to 0.05, *p* = 0.01). If these differences were maintained for 1 year, at the population level, this would result in 3,000 QALYs gained per 100,000 people.

In the base–case analysis, the ICER for the HIV screening arm versus SOC was US$901.29 per QALY gained, and the ICER for HIV-TB screening versus SOC was US$4,620.47 per

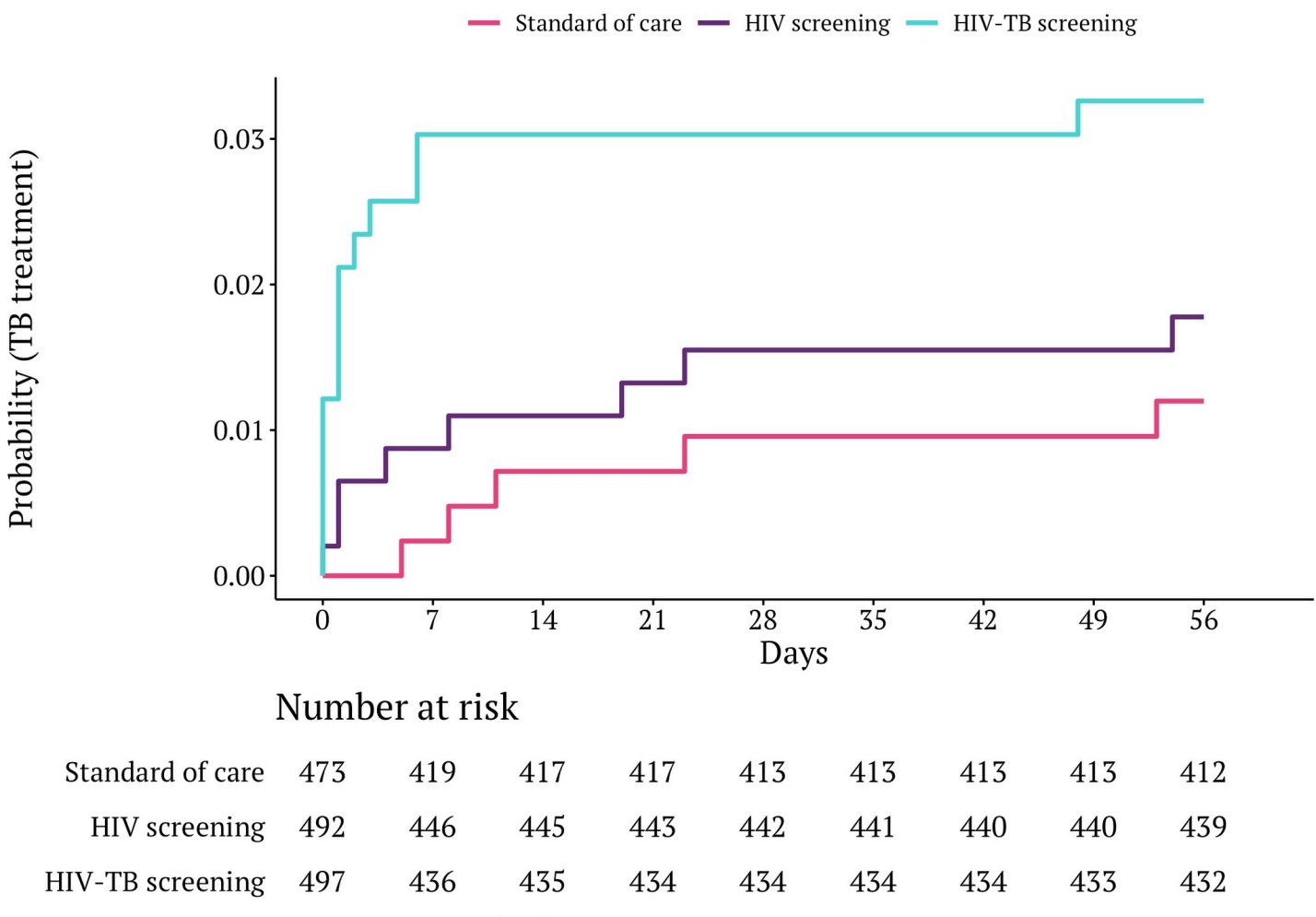

**Fig 3. Time to initiation of TB treatment by trial arm.** TB, tuberculosis.

QALY gained (Table 3). At the cost-effectiveness thresholds of US$400, US$800, and US$1,200 per QALY, the probability of cost-effectiveness for HIV screening was 3.0%, 36.2%, and 83.9%, respectively. Across all these 3 cost-effectiveness thresholds, the probability of cost-effectiveness for HIV-TB screening was 0%.

## Discussion

We have shown that the yield of HIV and TB among symptomatic adults attending primary care in Malawi was higher and that the time to TB treatment initiation was significantly shorter where participants received oral HIV testing plus DXCR-CAD TB screening with Xpert confirmation compared to clinician-directed TB/HIV screening. Although computer-aided TB diagnosis was not cost-effective over 8 weeks in the within-trial analysis, participant quality of life was significantly improved, and further economic analysis over longer time horizons is required. Universal offer of oral HIV testing was cost-effective and could offer substantial individual and public health benefits, even in settings like Malawi where UNAIDS 90–90–90 targets are close to being met [31]. If implemented at scale in primary care, DCXR-CAD combined with oral HIV screening has potential to improve diagnosis and treatment of symptomatic TB and HIV among persons presenting to the health facility with cough.

**Table 3. Health-related quality of life outcomes by treatment allocation.**

| | Total cost (mean/ SE) | Incremental cost (95% CrI)[†] | QALYs (mean/ SE) | Incremental QALYs (95% CrI)[†] | ICER[†] | Probability cost-effective at cost-effectiveness threshold: | | |
| --- | --- | --- | --- | --- | --- | --- | --- | --- |
| | | | | | | US$400/ QALY | US$ 800/ QALY | US$ 1,200/ QALY |
| Base–case analysis[#] | | | | | | | | |
| SOC | 21.45 (3.18) | | 0.001 (0.002) | | | | | |
| HIV screening | 24.29 (1.61) | 3.58 (1.70, 5.45) | 0.007 (0.002) | 0.004 (0.003, 0.005) | 901.29 | 0.030 | 0.362 | 0.839 |
| HIV-TB screening | 41.01 (1.17) | 19.92 (18.17, 21.68) | 0.007 (0.002) | 0.004 (0.003, 0.005) | 4,620.47 | 0 | 0 | 0 |
| Sensitivity analysis—Imputed data using UK Tarif to derive EQ-5D utility scores[#] | | | | | | | | |
| SOC | 21.45 (3.18) | | 0.001 (0.003) | | | | | |
| HIV screening | 24.29 (1.61) | 3.58 (1.70, 5.45) | 0.010 (0.002) | 0.005 (0.004, 0.007) | 714.69 | 0.062 | 0.652 | 0.959 |
| HIV-TB screening | 41.01 (1.17) | 19.92 (18.17, 21.68) | 0.009 (0.002) | 0.005 (0.004, 0.007) | 3,841.67 | 0 | 0 | 0 |
| Sensitivity analysis—Complete cases[#] | | | | | | | | |
| SOC | 20.19 (2.77) | | 0.001 (0.001) | | | | | |
| HIV screening | 23.72 (1.31) | 3.91 (−2.72, 10.53) | 0.007 (0.002) | 0.004 (0.0005, 0.008) | 953.85 | 0.266 | 0.445 | 0.595 |
| HIV-TB screening | 40.33 (0.91) | 19.94 (13.50, 26.38) | 0.007 (0.002) | 0.005 (0.001, 0.009) | 4,139.92 | 0.007 | 0.008 | 0.09 |
| Sensitivity analysis—Imputed data using lower cost for digital CXR (US$5)[*][#] | | | | | | | | |
| SOC | 21.11 (2.95) | | 0.001 (0.002) | | | | | |
| HIV screening | 24.16 (1.42) | 3.57 (1.72, 5.41) | 0.007 (0.002) | 0.004 (0.003, 0.005) | 961.12 | 0.020 | 0.292 | 0.748 |
| HIV-TB screening | 35.56 (1.09) | 14.53 (12.74, 16.32) | 0.007 (0.002) | 0.004 (0.003, 0.005) | 3,600.37 | 0 | 0 | 0 |

[†]Incremental estimates are in comparison to SOC arm.

[#]Adjusted for age, sex, marital status, highest level of education, employment status, and poverty quintile.

Bootstrapped differences (1,000 replications).

[*]In base–case analysis cost of digital CXR US$10.98.

CrI, credible interval; ICER, incremental cost-effectiveness ratio; QALY, quality-adjusted life year; SE: standard error, SOC, standard of care; TB, tuberculosis.

Diagnosis of TB in high HIV-TB burden and low-resource settings is extremely challenging. Under current recommendations, all clinic attendees should be asked about TB symptoms, interpreted according to HIV status, and followed by sputum testing with Xpert or microscopy of those with symptoms [6,32]. However, with nearly 60% of unselected acute care clinic attendees reporting TB symptoms [7], the current recommended screening approach for TB is not reliably implemented, reflecting workload, turnaround time, affordability, and throughput issues for sputum-based diagnostics. In the absence of an accurate, point-of-care diagnostic test for TB [33], alternative diagnostic algorithms that reduce demand for sputum-based diagnostics offer substantial potential benefits. For DCXR-CAD, these include minimal consumables, high patient throughput, and highly sensitive results available in minutes allowing large numbers to be screened for TB [34]. If DXCR-CAD accuracy is sufficiently high, there are potential cost-savings and infection control benefits from reducing workload and demand for sputum tests, as well as other clinical benefits from DCXR-CAD, including rapid screening for COVID-19 [35]. However, implementation of DCXR-CAD in high HIV/TB burden settings where they are most needed can be challenging and requires careful consideration by Ministries of Health, as well as technological optimisation for low-resource settings and operational research to support deployment. Particular challenges can include absent or intermittent power supplies, limited internet availability, availability of maintenance and servicing

personnel, and laboratory capacity to handle potential increased sputum specimen samples for confirmatory testing, although these were all overcome in this trial.

Here, we investigated "triage testing" adults with cough using automated DCXR-CAD screening accepting a CAD4TB score cutoff that provided high sensitivity (but correspondingly low specificity) combined with highly specific sputum Xpert confirmation. We achieved our aim of increasing speed and completeness of TB diagnosis in patients with cough of any duration but still required sputum from 61% (305/497) participants, a proportion that would have been similar to that in the SOC arm had national algorithms been consistently applied, emphasising the need to further optimise DCXR-CAD thresholds for confirmatory testing.

Potential approaches to further reducing sputum test demand include increasing the CAD4TB score used to define presumptive TB, or adding in a further rapid screening step such as point-of-care C-reactive protein, or clinical risk prediction scores [36,37]. These should be implemented and evaluated in conjunction with health system and laboratory strengthening. DCXR-CAD screening software is rapidly evolving, exceeding expert radiological reference standards across a range of target diseases and use-cases [35,38]. However, evidence for health impact or cost-effectiveness under route programmatic conditions—arguably of greatest importance to health planners—is extremely limited. Moreover, data from low-resource settings, where computer-aided systems could have greatest benefit in overcoming limited health worker capacity, are scarce. The CAD4TBv6 system has reported sensitivity of 91% and specificity of 84% compared to sputum Xpert [17], performing significantly better than expert radiologists and meeting target product profiles for a community TB screening test [39]. Important questions include the extent to which performance of DCXR-CAD systems varies by level of health system and patient characteristics such as sex, age, HIV status, disease stage, and epidemiological setting. As DCXR-CAD costs fall and systems develop, implementation research will still be needed to evaluate and optimise accuracy and ensure equity in access to care and health benefits.

Our effectiveness and cost-effectiveness estimates need to be considered in the context of low levels of investigation for HIV and TB seen in the SOC arm. For example, higher utilisation of routine clinic HIV testing and less accurate TB sputum screening in the SOC arm would reduce the likelihood that the HIV screening arm was cost-effective but may conversely increase the likelihood that offering more accurate TB screening with DCXR-CAD would be cost-effective. Our economic analysis demonstrates that optimisation of this TB triage testing approach may be possible. We are undertaking further modelling to evaluate the impact of lowering DCXR-CAD implementation and running costs; refinement of CAD thresholds to reduce the number of confirmatory sputum Xpert tests conducted (the major cost-driver here), while maintaining low false negativity rates in settings with varying pretest TB probabilities; alternative TB confirmatory tests that maintain high specificity and capability to be implemented in low-resource settings; and task-shifting to allow DCXR-CAD scale-up where radiographers are scarce or not available. Importantly, although CAD thresholds would ideally be adapted to local epidemiological characteristics and available resources, the large numbers of participants required and cost of undertaking diagnostic accuracy studies often precludes this, as in this trial. To minimise the need for multiple diagnostic accuracy studies with microbiological reference standards prior to implementation, modelling of accuracy at varying thresholds—as well as repeated surveillance and threshold adaption under programmatic conditions—will be required.

Malawi, similar to many countries in sub-Saharan Africa, has made tremendous progress in increasing access to HIV testing and treatment [21]. In this trial conducted in a busy urban neighbourhood of Blantyre, where adult HIV prevalence approaches 20% [21], greater than 90% of participants reported knowing their HIV status at recruitment, and >95% of HIV–

positive participants were taking ART. Modelling data suggests that where very high levels of population ART coverage are achieved, TB incidence will rapidly decline [40]. When planning this study, we assumed that 17% of participants would initiate TB treatment by 56 days based on data from a previous study [24]. However, we found TB treatment initiations and prevalence of microbiologically confirmed TB to be substantially lower than expected. We speculate, based on accelerated declines in Blantyre TB case notifications in our citywide TB surveillance system and from prevalence survey data, that public health interventions such as HIV testing and universal treatment, active case finding for TB, and scale-up of isoniazid preventive therapy may have caused this reduction. We did not estimate the diagnostic accuracy of DCXR-CAD, as baseline sputum samples were not collected from participants with a low CAD4TB score; if DCXR-CAD is widely implemented, surveillance of accuracy at DCXR-CAD thresholds using microbiological and clinical reference standards should be part of routine monitoring and evaluation activities. Nevertheless, the substantial individual and public health benefits offered by routine HIV testing of adults attending primary care with TB symptoms—even in a setting such as Malawi where HIV testing and treatment are extremely high—demonstrates that this should remain a central pillar of primary healthcare in countries with generalised HIV epidemics.

There were a number of limitations to this trial. In this pragmatic trial, alternative diagnoses to TB were not investigated, with participants referred to routine clinic or hospital care as required. If DCXR-CAD is widely implemented, clear clinical pathways for people who have an abnormal CAD4TB score, but who are sputum Xpert negative, will be required [41]. Although more participants initiated TB treatment in the HIV-TB screening arm compared to the other arms, the proportion with undiagnosed TB was similar across arms. It is not clear why this is the case but may be due to chance with small numbers of events here. We had planned subgroup analyses to investigate whether the effects of interventions differed by sex, an important determinant of delayed TB care-seeking. However, the fewer-than-expected number of events precluded this and also resulted in a loss of precision in treatment estimates. This means that our findings may not be robust to any differential missingness between groups. However, loss to follow-up was similar between arms, and characteristics of participants followed up were similar to those lost to follow-up. Some participants who were lost to follow-up may have died. We took extensive measures to limit contamination between arms (including fingerprint verification of all participants before randomisation, delivery of interventions, and at follow-up) and to avoid modifying health worker behaviour in the study clinic, but it is possible that the study presence in the clinic may have modified usual care practices. Future trials of DCXR-CAD may consider cluster randomisation at the clinic level to reduce these risks and to potentially increase the number of events observed. For safety reasons, remote radiologists (not usually available in similar primary care settings) provided a clinical interpretation of chest X-rays, although, given the very short median time to TB treatment initiation, this did not seem to influence TB treatment starts. Malawi has achieved very high levels of HIV testing and ART coverage; the effectiveness of these interventions should be evaluated in other settings.

In summary, DCXR-CAD with universal HIV testing for adults attending primary care with symptoms of TB increased TB diagnoses, shortened time to TB treatment, and was cost-effective at reducing undiagnosed HIV infection. If effective in larger trials and under programmatic conditions, this approach has potential to improve diagnosis and treatment of TB and HIV.

## Supporting information

**S1 Table. Characteristics of participants seen at day 56 visit compared to those lost to follow-up.**
(DOCX)

**S2 Table. Multiple imputation analysis of secondary outcomes with missing data.**
(DOCX)

**S1 Text. Protocol.**
(DOCX)

**S2 Text. Analysis plan.**
(DOCX)

**S3 Text. CONSORT Checklist.**
(PDF)

**S4 Text. Economic evaluation.**
(DOCX)

**S5 Text. CHEERS Checklist.**
(PDF)

## Author Contributions

**Conceptualization:** Peter MacPherson, Emily L. Webb, Elizabeth Joekes, David G. Lalloo, Titus H. Divala, Augustine T. Choko, Rachael M. Burke, Hendramoorthy Maheswaran, Madhukar Pai, S. Bertel Squire, Marriott Nliwasa, Elizabeth L. Corbett.

**Data curation:** Peter MacPherson, Emily L. Webb.

**Formal analysis:** Peter MacPherson, Emily L. Webb, Wala Kamchedzera, Hendramoorthy Maheswaran.

**Funding acquisition:** Peter MacPherson, Emily L. Webb, David G. Lalloo, Hendramoorthy Maheswaran, Madhukar Pai, S. Bertel Squire, Marriott Nliwasa, Elizabeth L. Corbett.

**Investigation:** Peter MacPherson, Wala Kamchedzera, Elizabeth Joekes, Gugu Mjoli.

**Methodology:** Peter MacPherson, Elizabeth Joekes.

**Supervision:** Elizabeth L. Corbett.

**Writing – original draft:** Peter MacPherson.

**Writing – review & editing:** Peter MacPherson, Emily L. Webb, Wala Kamchedzera, Elizabeth Joekes, Gugu Mjoli, David G. Lalloo, Titus H. Divala, Augustine T. Choko, Rachael M. Burke, Hendramoorthy Maheswaran, Madhukar Pai, S. Bertel Squire, Marriott Nliwasa, Elizabeth L. Corbett.

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
