## [Editor Report · Decision Letter 0]

7 Jan 2021

Dear Dr MacPherson, 

Thank you for submitting your manuscript entitled "Computer-aided X-ray screening for tuberculosis and HIV testing: a randomised trial and cost-effectiveness analysis in Malawi (PROSPECT)" for consideration by PLOS Medicine.

Your manuscript has now been evaluated by the PLOS Medicine editorial staff as well as by an academic editor with relevant expertise and I am writing to let you know that we would like to send your submission out for external peer review.

Kind regards,

Thomas J McBride, PhD

Senior Editor

PLOS Medicine

---

## [Decision Letter · Decision Letter 1]

12 Mar 2021

Dear Dr. MacPherson,

Thank you very much for submitting your manuscript "Computer-aided X-ray screening for tuberculosis and HIV testing: a randomised trial and cost-effectiveness analysis in Malawi (PROSPECT)" (PMEDICINE-D-20-05974R1) for consideration at PLOS Medicine. 

[LINK]

In light of these reviews, I am afraid that we will not be able to accept the manuscript for publication in the journal in its current form, but we would like to consider a revised version that addresses the reviewers' and editors' comments. Obviously we cannot make any decision about publication until we have seen the revised manuscript and your response, and we plan to seek re-review by one or more of the reviewers. 

We expect to receive your revised manuscript by Apr 02 2021 11:59PM. Please email us (plosmedicine@plos.org) if you have any questions or concerns.

We look forward to receiving your revised manuscript. 

Sincerely,

Dr Raffaella Bosurgi

Executive editor 

PLOS Medicine

plosmedicine.org

Comments from the reviewers:

Reviewer #1: Computer aided diagnostic systems have been evaluated in my multiple studies and have been shown to have utility in triaging for TB diagnosis. The finding that use of CAD in the screening reduces the time to diagnosis is not a surprise finding.

Specific comments

1. Previous studies evaluating CAD systems are not adequately referenced

2. There is not much description given of the HIV and TB programs in the setting. For example in most high TB and HIV settings, opt out HIV testing is offered to all patients accessing services. The study suggests that this is not offered in routine services and that the study was offering this outside of routine services. Is this correct?

3. CAD threshold setting is not fully described. How was the threshold of >=45 arrived at . The threshold setting is based on the prevalence of TB in the population being studied. Was this taken into consideration. Were there any patients with CAD score below 45 on whom Xpert was positive?

4. Since there is limited CXR accessibility in most TB and HIV high burden countries, the authors could have considered a HIV-TB screening arm ( without CXR) and and HIV-TB screening arm( with CXR) and compare the findings. TB screening is recommended for every HIV positive patient whenever in contact with the health services- generating evidence for the impact of on time to TB treatment, TB diagnosis can support the scale up of routine TB screening in HIV positive patients even when there is no CXR with CAD.

5. It would be helpful to report the the retention at day 56 instead of leaving it to the readers to calculate. If there were any drop outs, what was the reason for drop out ( or not returning at day 56)- If unknown or due to death can the assumption be made that some patients could have died from undiagnosed TB?

Reviewer #2: The manuscript by Macpherson et al from the Malawi research program supported by the Liverpool and London Schools of Hygiene and Tropical Medicine investigates the impact of an enhanced approach to TB diagnosis in early detection of TB in PLHIV and early TB treatment. A single clinic site, Bangwe Health Centre in Blantyre was chosen for the study. The diagnostic enhancement is a study staff controlled digital chest xray with an interpretation software application followed by sputum Gene Xpert probe for all xrays likely to be TB above a pre-set cutoff. The impact was compared with 'Standard of Care' managed by Ministry of Health staff and a second comparator group of study staff-managed 'oral HIV screening'. The main finding was median time to TB treatment which was 11 days for SoC, 6 days for oral HIV and 1 day for HIV and TB. The Hazard Ratio for HIV-TB compared with SoC is calculated at 2.84 which was statistically significant. A cost analysis is reported in a subsidiary section and found no chance of cost-effectiveness for the HIV-TB intervention although oral HIV screening was probably cost effective in the short term.

The primary audience for such studies should be Departments of Public Health, in particular in this case the Malawi Ministry of Health, but they may not find this study as helpful as it could have been. My main criticism is that, although the study is described as 'pragmatic', it is designed to measure the impact of complex and expensive pieces of technology to diagnose TB. A number of 'pragmatic' studies over the last 5 years of 'breakthrough' molecular diagnostics for TB, implemented in LMIC health systems, have been disappointing. One of the main reasons is that the systems - human resources, facilities, supply chains etc. - are not reliable enough to deliver the increased sensitivity and specificity of new technology. In this study, performed at a single site that accommodated all three arms, two of the arms were supervised by 'study staff' that have a large impact on implementation and adherence to standards of care, compared with the Ministry public health workers supervising the SoC arm. Thus there was benefit observed in the oral HIV arm supervised by study staff over the SoC arm, even though no direct TB diagnostic enhancement was used. It is not mentioned how much the study staff added to the cost of care but the impact of 'oral HIV' arm combined with its cost-effectiveness may have been more impressive, sustainable, and attractive to public servants than the high technology solution.

As far as the report in itself - it is well written, pretty clear, and the figures and analysis seem clear and supportable. Some specific criticisms - the digital xray platform and interpretation software is not well described. The study staff are not described - how many people, what level of training, location within Bangwe Clinic. A brief discussion of 'cross-contamination' between the study and SoC within a single clinic is included but I think this could be a major issue. Finally, Malawi generally has implemented an efficient national ART program with excellent coverage and adherence and this may well be responsible for unexpectedly low rates of community TB and present the most cost-effective approach. A discussion of the use of enhanced TB diagnosis in places where there is a much lower ART program efficiency should be included.

Reviewer #3: General:

The paper reports an innovative approach to try and improve diagnosis of both TB and HIV in a high HIV prevalence setting among people presenting to the clinic with cough. The study provides additional useful information on the importance of oral HIV self-testing, however, the importance of oral HIV self-testing in Blantyre and similar high HIV prevalence settings and OPD settings has been well described [1,2]. The use of Computer-aided X-ray screening for TB is innovative, but findings are limited to patients with cough presenting to the facility, which needs to be clear in the title. In addition, the combined HIV and TB screening intervention was not cost-effective. The paper provides important data points in this field, but the authors could and, in my opinion, should do more to describe the considerable cost, feasibility, and human resource challenges associated with rolling out Computer-aided X-ray screening in low-and middle income countries like Malawi where the burden of undiagnosed TB is highest.

Title

1. The study population represents symptomatic persons attending acute primary care clinics:

- The title could include the population studied such as: "Computer-aided X-ray screening for tuberculosis and HIV testing among adults presenting with cough: a randomised trial and cost-effectiveness analysis in Malawi (PROSPECT)"

This is important because a significant percentage of patients with active TB are asymptomatic [3]. In addition, the four-symptom TB screen is only able to detect 51% of active TB cases among PLHIV on ART [4]. Therefore, this is an intervention targeted at detecting symptomatic TB where the symptom is cough (and HIV infection among people with at least one symptom (cough))

Abstract:

2. The primary outcome was time to TB treatment. Is there a reason that the authors and investigators chose this as the primary outcome? Several studies have shown that even though new diagnostic approaches, like using Xpert instead of smear microscopy, have shortened the time to TB treatment, these approaches did not significantly improve patient important outcomes like morbidity or mortality [5].

3. One of the secondary outcomes is undiagnosed or untreated TB. Is this because the authors could not differentiate between undiagnosed and untreated TB based on available data in the medical records? If the authors could differentiate diagnosis from treatment, wouldn't undiagnosed TB be the cleaner outcome of interest because the intervention is designed to increase diagnosis of TB? Were there cases where an individual was diagnosed but not treated?

4. In the second last sentence of the abstract, the abstract talks about undiagnosed/untreated HIV. Again, can the authors indicate whether (1) it was impossible to differentiate between undiagnosed and untreated HIV, or (2) this represents a combined outcome. 

- In the abstract, the intervention itself seems focused on screening for HIV and no mention of the linkage component is made. However, on line 136: it suggests that support for ART enrollment was provided as part of the intervention - this is important for the abstract and should ideally be included. E.g., "oral HIV testing (HIV screening) and linkage to treatment for those who screen positive."

5. In the conclusions of the abstract, the term "universal HIV testing" is used, but earlier in the abstract the term HIV screening is used. Wouldn't "universal HIV screening" more accurately reflect the intervention? 

6. In reporting the primary outcome analysis, it is a little unclear which is the primary outcome comparison - is it the median time to TB treatment? The hazard ratio analysis compares rates of TB diagnosis between arms. Can the authors clarify why the statistical test used was Cox proportional hazards ratio analysis when the primary outcome comparison is of medians [6,7]? 

7. In the conclusion, does the term "completeness of diagnosis" mean completeness of both HIV and TB diagnosis? For clarity suggest adding the word "HIV and TB diagnosis".

Methods:

8. Page 8 - line 125 "we intended that the SOC arm would reflect the routine TB and HIV screening"

- To what extent is it truly possible to ensure that SOC stayed the same. Surely the healthcare workers would have been aware of the study and its hypothesis, and could this have affected SOC (i.e., improved quality) through a type of Hawthorne effect?

9. Page 9 - line 148: How many of the suspect TB cases were identified by the consultant radiographer rather than the CAD4TB?

- Most primary healthcare clinics will not have ready access to a consultant radiographer, so the authorship team should ideally specify whether this radiologist checking the x-ray results was part of the intervention or not?

10. Page 11 -Line 181-182: As noted in the abstract comments, please can the authors explain and clarify the terminology "undiagnosed/untreated tuberculosis" and "undiagnosed/untreated HIV". The key questions are: (1) do these represent composite outcomes? (2) were the authors able to differentiate between diagnosis and treatment? And (3) given the intervention on the TB side, which seems focused on diagnosis and not linkage to treatment, is it appropriate to include the TB treatment component in the secondary TB outcome? If linkage to TB treatment was part of the intervention, this should be clarified in the abstract and methods section. If someone was diagnosed with TB but not treated for TB, would this contribute to the numerator of the percentages reported for each arm?

- Similarly, if someone was diagnosed with HIV but not treated for HIV, would this contribute to the numerator of the percentages reported for each arm?

- What is the rationale behind the decision the authors made?

11. Page 12 - cost-effectiveness analysis:

- Please include the following details in the main manuscript text in the methods section - without this information, it is hard to understand what the authors did. Since word limits are not a constraint, the following text should be included: "Malawi does not have formal cost-effectiveness threshold. We therefore compared the estimated ICERs to WHO-recommended thresholds using the gross domestic product (GDP) per capita for the country. Interventions that have an incremental cost per gain in QALY less than the national GDP per capita were defined as "very cost-effective", and those less than three times GDP per capita as "cost effective"11. The GDP per capital of Malawi is approximately US400 per capital. We therefore used the bootstrapped replications to present the probability the two interventions (HIV screening; TB-HIV screening) were cost-effective at increasing cost-effectiveness thresholds: US$400/QALY; US$800/QALY; US$1200/QALY. The probability represents the proportion of the 1000 bootstrapped replications where the estimated ICER was below these cost-effectiveness thresholds."

Results:

12. The Fig 1 did not show up well on the manuscript I reviewed.

13. Can the authors add the percentages for those seen/vital status and the % lost to follow-up

14. Seems like close to 20% were lost to follow-up? Will need to address this potential source of selection bias and measurement error.

15. Line 219 - the problem of asymptomatic TB among the +-20% of clients living with HIV and mostly on ART needs to be considered in the discussion section.

16. Line 231 - 232: the CAD4TB approach screened in over two thirds of the population for GeneXpert (68%), but only 4% of the screened in population tested postive for microbiologically confirmed TB. A positive predictive value of only 4% among a group of symptomatic patients among whom HIV prevalence was 20%, seems low. Is the specificity of the CAD4TB approach too low? Do you have any data on TB prevalence among those who screened negative with the CAD4TB?

17. Table 2: Can the 95% CI of the number initiating TB treatment be added?

18. Line 276: the EuroQoL EQ5D utility score increases of 0.03 in both arms - this seems a small number. Is it possible to contextualize what a 0.03 means in the discussion section? 

19. The fact that the EuroQoL increases was 0.03 in both the HIV and HIV-TB arms, does this suggest that all the improvements were due to the HIV screening? Would the authors have expected a dose-response effect if additional improvements from undiagnosed TB on top of undiagnosed HIV were to be observed?

Discussion:

20. Page 23: Note: the line numbering is absent from the Discussion

21. Page 23: Laste sentence of first paragraph of the Discussion section: The following sentence should be revised to indicate that the intervention helped detect symptomatic TB, and helped detect HIV among those with at least one symptom "cough". In addition the words "rapidly and efficiently" should be removed. E.g., the sentence could be edited to read: "If implemented at scale in primary care, DCXRCAD combined with oral HIV screening has potential to improve diagnosis and treatment of symptomatic TB and HIV among persons presenting to the health facility with cough."

- The word "efficiently" needs to be deleted because (1) the paper did not assess the term "efficiency" from a costing perspective, and (2) from the cost-effectiveness analysis, the HIV-TB component was not cost-effective. If the HIV-TB screening component was not cost-effective, why would the authors consider it efficient? the word "rapidly" needs to be removed because the paper provided no evidence that either is shortened time for the patient at the clinic, or that DCXRCAD could be rapidly rolled out.

22. Page 23: The authors state: "However, with nearly 60% of unselected acute care clinic attendees reporting TB symptoms,[7] the current recommended screening approach for TB is not reliably implemented, reflecting workload, turnaround time, affordability and throughput issues for sputum-based diagnostics." The authors then list perceived advantages of the chest x-ray screening approach. However, the authors do not provide a ballanced view of the pros and cons of the chest radiography screening approach. There are substantial barriers facing widespread scale-up of the proposed intervention including:

(1) no abnormalities are definitive of TB and therefore specificity is low, 

(2) special equipment with constant source of electricity needed, 

(3) Equipment manintence is needed which is a challenge in LMIC,

(4) Trained personnel are needed to operate the equipment, 

The authors observed the intervention was not cost-effective in their own analysis.

The authors need to better describe the challenges associated with wide scale-up of the propsoed intervention, especially in a very resource-contrained country like Malawi where only 10% of the population has access to electricity and less than 60% of health facilities have any supply of electricity. Even those facilities that have a supply of electricity have routine, unscheduled blackouts.

23. Page 23: The authors note that all persons attending these clinics should be screened for HIV and TB risk, given the very high prevalence of HIV. For those clients with HIV, the WHO 4-symptom screen is recommended. Instead of introducing more complicated TB screening approaches with chest radiography and other needs (e.g., stable electricity, maintenance, skilled personnel), what would be the cost and impact of improving current WHO-recommended approaches to screening for TB among PLHIV and HIV-negative persons by providing additional healthcare workers, and training in current algorithms? In at least one study, this was effective among PLHIV [8]. There is only limited discussion of other approaches to improving screening for TB, and a broader acknowledgement of other potential approaches (e.g., health system strengthening to improve implementation of current algorithms, and use of simple risk scores) could ideally be mentioned. 

24. The authors state that CAD4TBv6 has high specificity (84%), without mentioning the population in which these data come from - can this be added? In addition, over 60% of persons attending the clinic with cough screened into Xpert screening after CAD4TBv6 screening with only 4% yield. Do the authors have any data on sensitivity and specificty of the TB screening approach taken in this study?

25. In the concluding sentence, the authors state: "If effective in larger trials and under programmatic conditions, this approach has potential to rapidly and efficiently improve diagnosis and treatment of TB and HIV.

- Recommend removing the words "rapidly and efficiently" because the data in this paper do not support this.

References

1. Choko AT, Corbett EL, Stallard N, Maheswaran H, Lepine A, Johnson CC, et al. HIV self-testing alone or with additional interventions, including financial incentives, and linkage to care or prevention among male partners of antenatal care clinic attendees in Malawi: An adaptive multi-arm, multi-stage cluster randomised trial. PLoS Med. 2019;16(1):e1002719. PubMed PMID: 30601823.

2. Dovel K, Shaba F, Offorjebe OA, Balakasi K, Nyirenda M, Phiri K, et al. Effect of facility-based HIV self-testing on uptake of testing among outpatients in Malawi: a cluster-randomised trial. The Lancet Global health. 2020;8(2):e276-e87. PubMed PMID: 31981557.

3. Drain PK, Bajema KL, Dowdy D, Dheda K, Naidoo K, Schumacher SG, et al. Incipient and Subclinical Tuberculosis: a Clinical Review of Early Stages and Progression of Infection. Clinical microbiology reviews. 2018;31(4). PubMed PMID: 30021818.

4. Hamada Y, Lujan J, Schenkel K, Ford N, Getahun H. Sensitivity and specificity of WHO's recommended four-symptom screening rule for tuberculosis in people living with HIV: a systematic review and meta-analysis. The lancet HIV. 2018;5(9):e515-e23. PubMed PMID: 30139576.

5. Auld AF, Fielding KL, Gupta-Wright A, Lawn SD. Xpert MTB/RIF - why the lack of morbidity and mortality impact in intervention trials? Trans R Soc Trop Med Hyg. 2016;110(8):432-44. PubMed PMID: 27638038.

6. Theron G, Zijenah L, Chanda D, Clowes P, Rachow A, Lesosky M, et al. Feasibility, accuracy, and clinical effect of point-of-care Xpert MTB/RIF testing for tuberculosis in primary-care settings in Africa: a multicentre, randomised, controlled trial. Lancet. 2014;383(9915):424-35. PubMed PMID: 24176144.

7. Spruance SL, Reid JE, Grace M, Samore M. Hazard ratio in clinical trials. Antimicrob Agents Chemother. 2004;48(8):2787-92. PubMed PMID: 15273082.

8. Auld AF, Agizew T, Mathoma A, Boyd R, Date A, Pals SL, et al. Effect of tuberculosis screening and retention interventions on early antiretroviral therapy mortality in Botswana: a stepped-wedge cluster randomized trial. BMC Med. 2020;18(1):19. PubMed PMID: 32041583.

Reviewer #4: MacPherson and colleagues describe the results of the PROSPECT trial in Malawi. The trial is both impressive and informative despite not having observed as many endpoints as it needed to be well-powered for the result. The manuscript is very well-written. I especially commend the many ways that the authors went above and beyond to make the manuscript transparent and clear. 

It is really helpful to have the power analysis described, to help readers explain why the trial seemed to be adequately powered at the outset but ended up having so few events in the primary outcome. 

The Appendix is a terrific addition, and having the unit cost assumptions broken out in great detail is an excellent addition to the literature and a leg up to future researchers.

Below are some suggestions for improving the manuscript.

Major points:

The article presents both HIV and TB outcomes, but the HIV component is presented as just a footnote. The Introduction only discusses TB, providing no context about HIV in this setting. To balance this, consider moving some of the HIV background (such as local HIV prevalence and ART coverage) from Discussion to Introduction and consider ways to balance the other sections of the paper.

There was a statistically significant reduction in undiagnosed HIV on on Day 56, but this finding does not seem to jive with the reported HIV diagnoses leading up to Day 56 (20, 25, and 15 new diagnoses in the SOC, HIV, and HIV-TB arms). How can one reconcile the finding of similar numbers diagnosed through Day 56 in each arm, and yet more undiagnosed cases of HIV on Day 56 in the SOC arm (10 compared to 2 and 1 in the other arms)? One would expect, if the arms were balanced in terms of number with undiagnosed HIV, that more HIV diagnoses would need to occur in the HIV and HIV-TB arm in order to "deplete" those arms of undiagnosed PLHIV. 

It is surprising that a majority of patients received a positive result from the triage test at the cut-off value used in this study, yet the actual prevalence of Xpert-positive TB was quite low, given the previously published CAD4TB performance of 91% sensitivity and 84% specificity. Could the authors claculate the specificity of the CAB4TB test that was observed in the trial? It would also be helpful if the Discussion could provide any insights into why the specificity might be so much lower than expected. 

Relatedly, the paper could provide more information about co-occurance of TB and HIV in the cohort. What was the HIV and ART status of those who screened positive with the CAB4TB test? Did CAD4TB specificity differ by HIV status? ART status? 

Do the investigators have access to the raw CAB4TB score (rather than just the binary outcome of whether or not it met the cutoff)? It could provide insight if the authors could tabulate (or show in a bar graph) the number of patients with different CAB4TB scores colored according to whether they had a positive Xpert, negative Xpert, or did not receive Xpert testing after the x-ray. Augmenting the table/graph with symptom and smear data stratified by CAB4TB score may also help inform the critical issue of specificity.

Cost-effectiveness analysis appears (at least based on what I understood from the Appendix) to only take into account the EuroQoL results, but in general one would expect that the main health benefit of prompt diagnosis would be later avoidance of critical illness and death, with only a small proportion of the effect being immediate quality of life after diagnosis. Is there a way to up-weight or forecast undiagnosed HIV and TB cases to account for future QALYs? Many published mathematical models make assumptions about future mortality risk for HIV and TB accounting for prompt vs delayed diagnosis; some of these models are open-source and available for download free of charge.

Minor points:

Abstract: 

The abstract uses the label "TB-HIV screening" but elsewhere in the manuscript it is referred to as "HIV-TB screening." Helpful to use consistent terminology.

"Undiagnosed/untreated" -- unclear if this means "undiagnosed OR untreated" or if it means "undiagnosed AND untreated." Please specify. If it is "OR" consider simply stating "undiagnosed" since all indiagnosed individuals would be untreated. 

This sentence is difficult to understand -- "HIV screening reduced the proportion with undiagnosed/untreated HIV..." 

Introduction:

"which can approach 60%" -- of all attendees?

Would be helpful to provide context and references/rationale for the CEA thresholds used.

Methods

The ART linkage intervention should be specified. "HIV-positive participants were supported to register for ART at the online clinic." -- what does the support entail? 

The method of recall of participants whose X-ray was flagged by the radiologist should also be specified.

Has the EuroQoL EQ5D score been validated in Chichewa? Please site validation studies.

It isn't clear why linear regression is used to analyze the change in EQ5D by arm. What is the independent variable?

Typo: "CD4TB" rather than "CAD4TB"

Results: 

What was the reason for the large drop-off in terms of screen-to-enroll ratio? 

Important to qualify claims like "participants in the HIV-TB screening arm were more likely to have initiated TB treatment..." since this was not a statistically significant trend.

Figure 1 (Trial Profile) did not render correctly in the PDF. What I'm albe to see (but these might be incorrect due to the rendering issue) is of n=8,236 screened, n=6,773 were "excluded" (why?), n=159 say "Multiple inclusion criteria not met" (what if only one is not met?), n=70 "did not consent" and n=1 "Withdrew before randomization." Need info on why the 6,773 were excluded and what happened with the remaining 1,224.

Figure 2 would be helpful to overlay statistics on the figure itself.

Discussion:

Would be helpful for Discussion to address why it might be that the rates of undiagnosed TB on day 56 were similar across arms, despite 2-3x more TB diagnosis and treatment between arms. Why didn't the intervention reduce prevalence of undiagnosed TB on day 56? For example, do you suspect much of the undiagnosed TB in the SOC arm may have self-resolved? Or treatment may have been unsuccessful in the HIV-TB arm? 

Typo: "target produce profiles" -> target product profiles

Reviewer #5: 

Thanks for the opportunity to review your manuscript. My role is as a statistical reviewer, so my review is focused on the study design, data, and analysis (and the presentation of these). I have put general comments about the manuscript first, followed by comments specific to a section of the manuscript (with line numbers).

This trial compares two treatments (oral HIV tests, and HIV test + computer aided chest x-rays) with usual care in adults with cough attending primary health services in Malawi. Two key strengths are the study is the large size of patients enrolled, and how the study was embedded within normal clinical practice. 

My first concern is about the final event rate (TB treatment initiated) in the study. The sample size for the study was based on 17% of patients initiating TB treatment, with a reasonably large effect size (HR of 1.5). The study recruited ~ 500 participants per arm (~1500) total and saw TB treatment in 28 participants (~2% initiated TB treatment). There is a 'significant'' difference between the HIV-TB screening arm and usual care, but the estimated upper limit of the 95% CI for this effect is 7.87. How should this effect be interpreted given this very high upper limit? A relatively small missing-not-random effect could diminish the difference very quickly - how robust is this main finding given the drop-out rate observed? The lower event rate is mentioned as a limitation of the study but this is only mentioned in the context of sub-group analyses not performed (I agree that this was a good decision). At the very least this should be discussed directly in the context of the assumed event rate in the sample size calculation.

The Cox method is a robust for large samples, but I am also concerned that the very small number of events makes this inappropriate with the data collected. There are proposed modification of the log-rank test that should be more robust in these situations (e.g. Mehrotra + Roth, Stats in Medicine 2001: doi: 10.1002/sim.854.) that would be appropriate to apply here to ensure the event-rate does not cause a bias in the hazard ratio.

Was the study prospectively registered? Was a SAP created? Having access to the SAP would help with the review process.

With regards to the main outcome, what is the interpretation of comparisons against the oral HIV test arm? Is this interpreted as a more 'active' control? 

L113. Was block randomisation used? One arm (497) has 24 more participants than one of the other study arms (473)

L183. What exact type of binomial models were these? i.e. what link function was used in the generalised linear model? 

L183. Three pairwise-comparisons between the three arms was made - was any adjustment for multiplicity performed?

L187. How was this adjustment made? By change score or including baseline value as a covariate? 

L187. It looks like approximately 256 participants in total were able to be contacted at day 56. How was this missing data treated? Were these censored in the main analysis or were they excluded? 

L187. How were the assumptions in the models checked (e.g. PH assumption for Cox model, distribution of residuals in linear regression?)

[LINK]

---

## [Decision Letter · Decision Letter 2]

21 Jul 2021

Dear Dr. MacPherson,

Thank you very much for re-submitting your manuscript "Computer-aided X-ray screening for tuberculosis and HIV testing among adults with cough: a randomised trial and cost-effectiveness analysis in Malawi (PROSPECT)" (PMEDICINE-D-20-05974R2) for review by PLOS Medicine.

I have discussed the paper with my colleagues and the academic editor and it was also seen again by five reviewers. I am pleased to say that provided the remaining editorial and production issues are dealt with we are expecting to be able to accept the paper for publication in the journal.

[LINK]

We look forward to receiving the revised manuscript.

Sincerely,

Callam Davidson, 

Associate Editor 

PLOS Medicine

plosmedicine.org

Requests from Editors:

Please update your title to ‘Computer-aided X-ray screening for tuberculosis and HIV testing among adults with cough in Malawi (the PROSPECT study): a randomised trial and cost-effectiveness analysis’.

Abstract Methods and Findings:

* Please ensure that all numbers presented in the abstract are present and identical to numbers presented in the main manuscript text (e.g. line 52 ‘83%’ is 83.9% on line 321).

* Line 38: Please either update to ‘Secondary outcomes included’ if opting to only include the main outcomes, or list and report all of your secondary outcomes.

* Please specify who was blinded/masked.

* Please state that analysis was intention to treat.

* Please provide the number of participants lost to follow up in each group.

Please provide titles and legends for all figures (including those in Supporting Information files).

Please provide the unadjusted comparisons as well as the adjusted comparisons in Table 2.

Please update the CONSORT checklist to use section and paragraph numbers, rather than page numbers. Please add the following statement, or similar, to the Methods: "This study is reported as per the Consolidated Standards of Reporting Trials (CONSORT) guideline (S1 Checklist)."

Per CONSORT guidelines, please describe how investigator masking was maintained until after statistical analysis. Please also define "lost to follow-up" as used in this study. Other reasons for exclusion should be defined. Please reference these when updating the CONSORT checklist. 

Please report your cost-effectiveness analysis according to the CHEERS checklist provided at https://www.equator-network.org/wp-content/uploads/2013/04/Revised-CHEERS-Checklist-Oct13.pdf. Please provide a copy of the completed checklist as a supplementary file and reference this in the Methods (and when completing, use section and paragraph numbers rather than page numbers).

Please include the study protocol document, with any amendments, as Supporting Information to be published with the manuscript if accepted.

On line 312, please clarify what you mean by "significantly". If statistical significance is intended, please provide the relevant p-value. 

Citations should be in square brackets, and preceding punctuation.

Please remove the ‘Role of the funding source and data availability’ section from your methods – in the event of publication this information will be published as metadata based on your responses to the submission form.

Comments from Reviewers:

Reviewer #1: The use and role of digital CXR with CAD systems has previously been evaluated and shown to have utility in reducing time to diagnosis and increasing TB diagnosis. This work presented in this manuscript further adds to this existing body evidence though unfortunately does not adequately reference previous reports. 

However, the manuscript is well written and easy to follow and the authors have adequately responded to previous reviewer comments

Some limitations/ criticisms

1. The enhanced arms (HIV screening and HIV-TB screening )depended on trained study staff whilst the SOC depended on routine ministry of health staff. This limitation was pointed out a previous reviewer. The outcome of the enhanced arms could have been influenced by use of dedicated study staff. Dedicated staff should have been used for SOC arm too 

2. When done well, symptom screening can result in increased TB case detection in PLHIV, as previously pointed by another reviewer, the study should have included this with dedicated staff to provided the opportunity for comparison with CAD TB screening. This is especially important in settings that will in the foreseeable future still have very limited access to digital x-rays. CAD systems also come at a cost and thus we do not expect increased access to CAD systems immediately despite the new recommendations from WHO

3. Did calculations of cost effectiveness take into consideration the lifespan of digital x-ray machines? By how much did use of CXR reduce the number needed to test with Xpert and what was the cost savings on Xpert cartridges?

4. Ascertainment of mortality and lost to follow up is not well described in the methods, were participants asked to visit the facility at day 56? or did they come through at predefined follow up time points? And those lost to follow up, any effort made to determine reason, could some be moralities? 

5. Were any of the mortalities among those who were initiated on TB treatment? If so , though numbers are small, were they any differences that might indicate some differences in the different arms?

Reviewer #2: I have read the detailed responses of Dr MacPherson on behalf of his co-authors to the 5 reviewers of this submission. I am satisfied with the responses which show an appreciation of the issues raised by the reviewers and the limitations of the single-site Bangwe Clinic study. The results and conclusions of this necessarily limited study remain valid and the need for a larger and more robust study design in settings with different epidemiological and health system characteristics has been recognized.I think the updated manuscript has addressed the reviewers comments and and can be published without further revision.

Reviewer #3: Thanks for the opportunity to review the revision. Thanks to the authors for making many of the suggested changes. I have no additional comments.

Reviewer #4: The authors have addressed the reviewer's comments well. Just minor follow-up:

Reviewer #4 items 5 and 6: The authors requested guidance about whether to icnlude in the paper the additional data on HIV and ART status by CAB4TB score, as well as the new figure showing the distribution of numeric scores by important strata. These do seem appropriate to incorporate into the manuscript. The figure should use bar graphs or line graphs showing the number or proportion of participants with each score. It is difficult to ascertain the density from the jittered dots in the graphs provided.

Reviewer #4 item 15: Please cite the 2012 validation study of the EuroQoL in Chichewa.

Reviewer #4 item 19: Please qualify with something like "but this trend was not statistically significant."

Reviewer #5: Thanks for the revised manuscript. Your manuscript had had a thorough set of reviews and I appreciate the thoughtful replies to my original queries. Most of my queries have been resolved - the only things outstanding were some additional planned sensitivity analysis (from the SAP) that should be presented, along with some clarification about interim visits (see further down).

I agree that the Cox model appears to be the best option here - the low event rate is acknowledged completely in the discussion and thank for pointing me to the simulation study on event rates, that was helpful.

Unrestricted randomisation is fine - I would just add a few words as a qualifier in the info about randomisation (e.g. L123).

The thee pairwise comparisons are ok as well - 3 pre-planned comparisons is perfectly reasonable, but I would just add a clarifying phrase to the methods stating that there was no adjustment for multiplicity.

Apologies for not picking up the SAP in the first round of reviews, this for the most part matches the analysis described in the manuscript. The main difference I noted was that there was a planned sensitivity analysis using MICE for missing outcomes was planned, but these aren't reported. This would provide reassurance that missing data (at least under the missing at random assumption) does not affect the main results.

Going from p-values in the Supp table, it may look as though there are some key differences between the complete-case data and those lost to follow-up. Education/literacy and age are moderately different but the p-values reflect the sample size here more than any strong differences. A HIV-positive status was less likely in those lost to follow-up, but again this was not strong (13% vs 20%). Presenting the planned sensitivity analysis showing similar effects as the main analysis along with this descriptive data would settle the issue about missing outcome data in this study. Strictly speaking, censoring or excluding participants with missing outcome data means the analysis can't be considered 'intention-to-treat'.

For those who didn't make the Day 56 outcome assessment visit, what was the average (mean, median) length of follow-up? I wasn't clear from the description if there were interim visits between baseline and the Day 56 visit.

[LINK]

---

## [Editor Report · Decision Letter 3]

3 Aug 2021

Dear Dr MacPherson, 

On behalf of my colleagues and the Academic Editor, Dr Ruanne Barnabas, I am pleased to inform you that we have agreed to publish your manuscript "Computer-aided X-ray screening for tuberculosis and HIV testing among adults with cough in Malawi (the PROSPECT study): a randomised trial and cost-effectiveness analysis" (PMEDICINE-D-20-05974R3) in PLOS Medicine.

Please also make the following changes when updating your manuscript for submission:

* Line 281: At the end of the ‘Cost-effectiveness analysis’ subsection in the Methods, please add the sentence ‘This analysis is reported as per the Consolidated Health Economic Evaluation Reporting Standards (CHEERS) guideline’ and reference the appropriate supplementary checklist in parentheses.

* Line 56: Please define the abbreviation QALY. 

* Line 56: Please update ‘84%’ to ’83.9%’ for consistency with the main text.

* Lines 90 and 99: Please remove the word ‘significantly’.

* Line 139: Please add ‘The’ before WHO.

* Line 185: Remove the instance of the word ‘routine’ as this appears to be erroneous. 

* Line 455: Update ‘scare’ to ‘scarce’

PRESS

Sincerely, 

Callam Davidson 

Associate Editor

PLOS Medicine

cdavidson@plos.org